# Optimal Timing and Treatment Modalities of Arytenoid Dislocation and Subluxation: A Meta-Analysis

**DOI:** 10.3390/medicina61010092

**Published:** 2025-01-08

**Authors:** Andrea Frosolini, Valeria Caragli, Giulio Badin, Leonardo Franz, Patrizia Bartolotta, Andrea Lovato, Luca Vedovelli, Elisabetta Genovese, Cosimo de Filippis, Gino Marioni

**Affiliations:** 1Maxillofacial Surgery Unit, Department of Medical Biotechnologies, University of Siena, 53100 Siena, Italy; 2Otorhinolaryngology-Head and Neck Surgery, Audiology Program, Department of Diagnostic Clinical and Public Health, University of Modena and Reggio Emilia, 41125 Modena, Italy; valeria.caragli@unimore.it (V.C.); elisabetta.genovese@unimore.it (E.G.); 3Otolaryngology Section, Department of Neuroscience DNS, University of Padova, 35100 Padova, Italy; badingiulio@gmail.com; 4Phoniatrics and Audiology Unit, Department of Neuroscience DNS, University of Padova, 31100 Treviso, Italy; leonardo.franz@unipd.it (L.F.); cosimo.defilippis@unipd.it (C.d.F.); gino.marioni@unipd.it (G.M.); 5Unit of Biostatistics, Epidemiology, and Public Health, Department of Cardiac, Thoracic, Vascular Sciences, and Public Health, University of Padova, 35100 Padova, Italy; patrizia.bartolotta@ubep.unipd.it (P.B.); luca.vedovelli@ubep.unipd.it (L.V.); 6Otorhinolaryngology Unit, Department of Surgical Specialties, Vicenza Civil Hospital, 36100 Vicenza, Italy; andrea.lovato.3@hotmail.it

**Keywords:** arytenoid dislocation, arytenoid subluxation, treatment, local anesthesia, general anesthesia, time to treatment

## Abstract

*Background and Objective*: Arytenoid dislocation (AD) and subluxation (AS) impact vocal fold mobility, potentially affecting the quality of life. Their management, including the timing and modality of treatment, remains a subject of research. Our primary objective was to assess and compare the available treatment strategies for AS and AD. *Material and methods*: the protocol was registered on PROSPERO (CRD42023407521). Manuscripts retrieved from a previously published systematic review were evaluated. To comprehensively cover the last 25 years, an updated literature search was conducted, screening PubMed, Scopus, and Cochrane databases. Review Methods: We included studies that reported treatment modalities and the time to treatment (TT) for AS/AD, with outcomes objectively evaluated. Data on treatment success were pooled, and the impact of TT on recovery outcomes was analyzed. *Results*: Thirteen studies involving 361 patients were included. The majority of cases were attributed to iatrogenic trauma following intubation. Closed reduction (CR) was the primary treatment, with high success rates for both general (success rate: 77%, CI: 62–87%) and local anesthesia (success rate: 89%, CI: 70–97%). The standardized mean difference for the TT effect on treatment outcome was −1.24 (CI: −2.20 to −0.29). *Conclusions*: The absence of randomized controlled trials and the overall moderate-to-low quality of the studies highlighted the importance of the finding’s careful interpretation. This meta-analysis underscores the effectiveness of CR in managing AS/AD, with both general and local anesthesia yielding high success rates. The findings highlight the importance of TT, suggesting that early intervention is paramount. Future clinical research is needed to further refine these findings and optimize treatment protocols.

## 1. Introduction

The cricoarytenoid joint is included within a complex muscular and ligament system, and it is characterized by an elaborate three-dimensional movement pattern, the biomechanics of which have been effectively studied through 3D modelling and vector analysis [1,2]. The cricoarytenoid joint takes on a central role in the laryngeal function as it regulates the position, tension and volume of the vocal cords, allowing the protection of the airways, cough, ventilation control, and phonation [1]. Anatomically, it is a diarthrodial joint, supported by a wide joint capsule and strengthened posteriorly by the cricoarytenoid ligament [3].

Arytenoid dislocation (AD) and subluxation (AS) are defined as a complete or partial loss of contact between the cartilaginous surfaces, respectively [4]. It is an uncommon pathological condition with iatrogenic trauma due to intubation being the most frequent etiology (the reported percentage of AS/AD due to intubation ranges from 77.8% to 87.6%), followed by external blunt trauma (12.4% to 15.9% of cases) while spontaneous AS/AD is rare [5]. More precise estimates of the incidence of arytenoid dislocation following endotracheal intubation have been recently provided, reporting a pooled incidence rate of 0.093% (95% CI: 0.045–0.14%) according to a meta-analysis [6]. However, the reported incidence varies significantly across studies, likely due to differences in the population studied, diagnostic criteria, and underdiagnosis in clinical practice. The identified risk factors include anemia, laryngomalacia, acromegaly, chronic steroid use, low BMI, and conditions such as Marfan syndrome, renal failure, GERD, and CHARGE syndrome [5]. Moreover, procedural factors during anesthesia and surgery may increase AS/AD risk: cardiovascular surgery has been associated with an incidence of 0.07%, which is potentially linked to transesophageal echocardiography, while bariatric surgery involving a calibrating oro-gastric tube showed a postoperative incidence of 0.8% [6]. The presence of debilitating symptoms, including hoarseness, dysphonia, dysphagia, and dyspnoea, requires the active collaboration of the patient and a need for specific diagnostic tools and specialist consultation (i.e., an otolaryngologist or a phoniatrician) [7]. As mentioned above, given that the most frequent etiology is iatrogenic following intubation [5], in some cases, the diagnosis could be delayed due to difficulty in communicating with the patient during intensive care unit hospitalization [7]. Although cases of spontaneous resolution have been documented, this clinical situation usually requires adequate and timely treatment, as it could lead to a hypermobile joint or ankylosis resulting from hemarthrosis [8,9]. The most widely used treatment of AS/AD is closed reduction (CR) under general anesthesia. More recently, CR under local anesthesia, first described in 1978, has also become established [10]. Both techniques have shown good success rates [5], albeit local anesthesia techniques more frequently need to be repeated to achieve reduction than the technique under general anesthesia [11]. A single-centre retrospective study conducted on 35 patients concluded that the appropriate time window to perform CR under local anesthesia was between the 13th and 26th day after arytenoid dislocation [12]. It was assumed that premature treatment might not be effective because inflammation, pain, mucosal swelling, and hematoma in the early period of injury could hamper the surgical technique, while delayed treatment could be limited by cartilage fibrosis and joint stiffness [12]. In the largest series reported in the literature about AS/AD, accounting for 63 patients treated in a single institution with CR under general anesthesia, patients with complete voice improvement received earlier treatment (average of 14.3 weeks), while those with significant and slight improvement had longer times to treatment (averages of 97.2 weeks and 279.9 weeks, respectively).

Recently, our group undertook an in-depth examination of the literature related to the prevalence, symptoms, and diagnosis of AS/AD. Nevertheless, we did not focus on a detailed qualitative and quantitative analysis of the available treatment methods [5]. The main purpose of the present updated systematic review and meta-analysis was to provide a comprehensive overview of the current treatment options for AS and AD, highlighting the most effective treatment for managing this uncommon condition. A secondary aim was to investigate the importance of time to treatment (TT) in influencing AS and AD outcomes and prognoses.

## 2. Materials and Methods

### 2.1. Protocol Registration and Research Question

The protocol for the present systematic review and meta-analysis was registered on PROSPERO (study ID CRD42023407521) in February 2023, and it is available in its full version at https://www.crd.york.ac.uk/prospero/ (accessed on 5 January 2025).

The research question of the present review was twofold: Is the recently proposed closed reduction under local anesthesia significantly more effective for treating AS/AD compared to closed reduction under general anesthesia? Additionally, is the interval between the diagnosis and treatment a valid prognostic factor? Therefore, according to the Population, Intervention, Comparison, Outcomes (PICO) criteria, this study focused on the following: (i) a population of patients with a clinically confirmed diagnosis of AS/AD; (ii) the intervention of closed reduction; (iii) a comparison if reported, but not mandatory; and (iv) outcomes defined as the successful relocation of the arytenoid—observed via video laryngoscopy—tacking into account TT as a prognostic factor.

### 2.2. Eligibility Criteria

Studies were included when the following general criteria were met: (i) original articles including clinically confirmed cases of AS or AD with detailed information about its diagnostic workout; (ii) the availability of information about the treatment modality and TT; and (iii) defined outcomes in terms of AS or AD reduction confirmed by video laryngoscopy and/or computerized tomography. Exclusion criteria were as follows: (i) study design in the form of a case report, editorial, letter to the editor, or review; (ii) a non-English language study.

### 2.3. Information Sources—Electronic Database Search

Electronic databases were searched for papers about AS/AD over the last 25 years. The first part of the research (from 1 December 1999 to 1 December 2019) was covered in the previous publication [5], while updated research regarded papers published from 1 December 2019 to 1 August 2024 on PubMed, Scopus and Cochrane. The search terms were “arytenoid dislocation” and “arytenoid subluxation”, separated by the Boolean operator “OR”. The “Related articles” option on the PubMed homepage was also considered. Titles and abstracts of papers available in the English language were examined by three authors (V.C., B.G., and A.F.). The identified full texts were screened for original data, and the related references were retrieved and checked manually for other relevant studies.

### 2.4. Data Extraction and Qualitative Assessment

Three authors (V.C., B.G., and A.F.) performed the data extraction and qualitative synthesis of the included studies. Once the studies had been retrieved from the electronic database search, a thorough evaluation against our predefined inclusion and exclusion criteria was performed to confirm each study’s adequacy for inclusion in our review and their relevance. Selection, comparability, and reporting outcome measures were evaluated according to the Newcastle–Ottawa Scale (NOS) [5]. No restriction about the minimum number of studies or level of consistency required for synthesis was applied. For each included manuscript, available information regarding the study design, population, condition (the description of laryngeal features), time from diagnosis to intervention, intervention (method of reduction), outcome and follow-up controls was extracted and stored in an Excel file (Microsoft Corporation, Redmond, WA, USA). Any disagreements were resolved by a discussion among the research team. Appropriate tables were created to summarize the results.

### 2.5. Quantitative Analysis

A meta-analysis approach was applied to compare the results of different arytenoid reduction approaches and the effects of TT on outcomes. Pooled weighted mean differences (WMDs) were estimated using a random-effects model.

A random-effects model was fitted to the data to consider both within- and between-study variability. The amount of heterogeneity, represented by tau^2^, was estimated using the restricted maximum-likelihood estimator. The Q-test for heterogeneity and the I^2^ statistic were also calculated to assess the presence and magnitude of heterogeneity among the studies. If any amount of heterogeneity was detected (tau^2^ > 0), a prediction interval for the true outcomes was provided. The Meta-Analysis Online Software 2024 (available at https://metaanalysisonline.com/) was used to perform the quantitative analysis.

## 3. Results

### 3.1. Retrieved Studies and Quality Assessment

A total of 61 papers was found through title and abstract screening: 20 studies were already retrieved [8,11,12,13,14,15,16,17,18,19,20,21,22,23,24,25,26,27,28] from the previously published systematic review, while 41 records resulted from the updated literature search of English-language studies through PubMed, Cochrane and Scopus databases (after duplicates were removed). After full-text screening, 48 articles were excluded because they did not match the inclusion criteria. Thirteen studies [11,12,14,15,16,17,20,22,24,25,26,27,28] were considered eligible for entering our systematic review. Figure 1 summarizes the study selection process.

All included studies had adequate relevance to the subject of this review. One investigation was prospective [28] and twelve were retrospective [11,12,14,15,16,17,20,22,24,25,26,28]. According to the NOS, 69.2% of the selected studies scored 5/8 [14,15,16,20,22,23,24,27,28], while 31.8% of them scored 4/8 [11,12,17,26].

### 3.2. Main Features of Included Studies

A total of 361 patients were enrolled; the mean age was 46.95 years, and 53.7% of them were male, as depicted in Table 1.

The most frequent etiology of AS/AD was iatrogenic trauma due to intubation (ranging between 61.76% [20] and 100% [18,22,24,25,28] of the cases), while the second reported cause was laryngeal external blunt trauma (from 15.9% [14] to 28% [28]). A few cases of spontaneous AS/AD were reported [15,16]. Sometimes, AS/AD causes were idiopathic or not recorded [12,14,20]. AS/AD’s most commonly reported symptom was dysphonia [11,12,14,16,17,20,22,24,25,26,27], followed by dysphagia [14,17,24], and dyspepsia [12,26].

In 50.1% of cases, arytenoids were found to be dislocated on the left side and in 31.6% on the right side. Bilateral arytenoid dislocation (AD) was disclosed in 2.2% of cases. The side of dislocation was unspecified in 16.1% of cases [16,27]. Various classification systems have been reported to categorize the direction of dislocation, including (i) the anterior–posterior classification [11,12,26,27]; (ii) a combined anterior–posterior/medial–lateral classification [16,17,25], which also considers subtypes based on the position of the arytenoid tip [16]; (iii) the anterior–posterior-complex classification [14]; and (iv) the cranio-caudal/medial–lateral classification [15]. Instead, the specific direction of dislocation was not defined by three groups [20,22,24,28].

Fibroscopy was used in 12 out of 13 studies to objectively assess AD/AS. In order to confirm the diagnosis, a computerized tomography scan was performed in 11 out of 13 studies, using high-resolution images and/or 3D reconstructions [11,12,14,15,17,18,20,22,25,26,27,28]. Sometimes, laryngeal electromyography (LEMG) was also used [14,15,16,25,27].

### 3.3. Treatment and Outcomes

Spontaneous recovery was found in nine AD/AS patients [12,14,15,20,22,24,27]; for all other cases, CR was a first-choice treatment. Rubin [14], Lou [22], Zheng [26], and Cao [20] reported having had to repeat the procedure more than once in refracted patients. In a small percentage of cases (33 out of 361, 9.15%), other surgical treatments (i.e., injection laryngoplasty, botulinum injection, and thyroplasty) were performed alongside CR in order to treat refractory cases, as depicted in Table 1 [14,15,20,27]. TT ranged from 1 to 6223 days, with a mean of 80.9 days (SD 122.9). After treatment, fibroscopy was performed to investigate vocal fold mobility, while voice quality was analyzed through the Voice Handicap Index (VHI), GRBAS scale, and acoustic voice analysis in the majority of studies [11,12,15,20,25,26,27,28]. Otherwise, direct questions to patients and/or subjective voice quality questionnaires were administered [11,12]. As depicted in Table 1, voice outcomes were as follows: 68.1% normal, 18.4% improved, 2.7% slightly improved, and 10.8% unchanged. Based on these categories, data were divided into two groups: successful treatment (comprising both normal and improved outcomes) and unsuccessful treatment (including slightly improved and unchanged outcomes). A meta-analysis was then conducted on these dichotomized data, with the findings detailed in the following section.

**Table 1 medicina-61-00092-t001:** Characteristics and main findings of included studies.

Reference	No. Cases (Sex)	Mean Age in Years ± SD (Range)	Type of AS/AD (No. Cases)	Etiology (No. Cases)	Instrumental Tests	Type of Treatment (No. Cases)	Mean TT in Days ± SD (Range)	Outcome (No. Cases)
Rubin 2005 [14]	63 (24 M, 39 F)	42.5 (18.6)	AD-L (35), AD-R (25) AD-B (3) ant (17) post (32) complex (5) ant -post (3) NR (6)	Intubation (49) External blunt trauma (10) Anesthesia (1) Whiplash (1) Idiopathic (1) NR (1)	Fibro CT LEMG Subjective and objective evaluation of voice quality	SR (2), CR (35), CR + injection laryngoplasty (21), CR + Thyroplasty I (2), Voice therapy (3)	Successful 680.4 ± 1196.3 Unsuccessful 1959.3 ± 2954	N (10) I (31) SI (9) U (5) Lost F-U (6)
Hiramatsu 2010 [15]	12 (8M, 4F)	65 (39–88)	AD-L (8), AD-R (4) Medio-caudal (6) medio-cranial (5) Latero-cranial (1)	Intubation (8) Idiopathic (4)	CT LEMG MPT AVA	SR (2) CR (3) Thyroplasty I (3) Untreated (4)	5.5 ± 6.6	I (3) U (3) Lost F-U (4)
Leelamanit 2012 [16]	29 (8M, 21F)	42.48 (12–77)	AD ant-med (28) AD post-lat (1) Subtypes I (18); II (6); III (4)	Intubation (26) Spontaneous (3)	Fibro LEMG Voice quality of life Patients’ global voice quality	CR (29)	5.45 ± 8.26	N (21) I (5) U (3)
Lee 2013 [17]	11 (7M, 4F)	55	AD-L (8) AD-R (3) ant-med (10) post-lat (1)	Intubation (9) External blunt trauma (2)	Fibro CT Subjective and objective evaluation of voice quality	CR (11)	49.8 ± 41.48	N (6) I (5)
Lee 2014 [27]	22 (10M,12F)	36.6	AD ant (16) AD post (6)	Intubation (16) External laryngeal trauma (6)	Fibro CT LEMG AVA VHI	CR (16), SR (1), CR + injection laryngoplasty (4), CR + botulinum (1)	21 (7–6223)	N (18) U (3)
Teng 2014 [11]	12 (8M, 4F)	35 (22–53)	AD-L (5) AD-R (6) AD-B (1) Ant (11) L-ant R-post (1)	Mechanical pressing (3) Traffic accident (6) Punch (2) Stick fighting (1)	Fibro CT VHI MTP	CR (12)	Successful 43.4 ± 34.1 Unsuccessful 157.7 ± 76.1	N (5) I (4) U (3)
Lee 2015 [28]	13 (6M, 7F)	54.1 ± 15.9	AD-L (8) AD-R (5)	Intubation	Fibro CT Subjective and objective evaluation of voice quality	CR (13)	22.2 ± 28.8	N (13)
Lou 2016 [12]	35 (21M, 14F)	51.7	AD-L (19) AD-R (16) ant (32) post (3)	Tracheal intubation for surgery (30) Ventilation (3) Gastric tube (2)	Fibro CT VHI MPT	SR (1) CR (34)	17.4 ± 11.7	N (34)
Cao 2016 [20]	34 (24M, 10F)	46.96 ± 18.85	AD-L (20) AD-R (12) AD-B (2)	Intubation (21) Blunt trauma (11) LMA insertion (2)	Fibro CT LEMG GRBAS VHI AVA	SR (1) CR (31) CR + injection laryngoplasty (2)	Successful 18.5 ± 9.5 Unsuccessful 41.9 ± 29.3	N (26) SI (7)
Lou 2017 [22]	28 (18M, 10F)	55 ± 12 (22–76)	AD-L (16) AD-R (12)	Intubation	Fibro CT VHI MPT	SR (1) CR (27)	NR	N (27)
Hung 2018 [24]	14 (6M, 8F)	36.9	AD-L (13) AD-R (1)	Intubation	Fibro	SR (1) CR (13)	Successful 6.2 ± 4.3 Unsuccessful 19 ± 24.1	N (11) U (2)
Wu 2019 [25]	57 (38M, 19F)	47.09 ± 17.82 (15–83)	AD-L (36) AD-R (19) AD-B (2) ant-med (53) post-lat (4)	Intubation (42) Blunt trauma (15)	Fibro CT LEMG VHI GRBAS	CR (57)	NR	N (24) I (15) U (18)
Zheng 2019 [26]	31 (16M, 15F)	50.97 (22–76)	AD-L (18) AD-R (13) ant (22) post (9)	Intubation (31)	Fibro CT VHI GRBAS AVA	CR (31)	20.2 ± 4.19	N (31)

Abbreviations: AD: arytenoid dislocation, AS: arytenoid subluxation; B: bilateral; AVA: acoustic voice analysis; B: bilateral, CR: close reduction, CT: computed tomography, Fibro: fibroscopy, F-U: follow-up, GRBAS: grave, roughness, breathiness, asthenia, strain, HNR: harmonics-to-noise ratio, I: improved, L: left side, MPT: maximum phonation time, N: normal/fully recovered, NOS: Newcastle–Ottawa Scale, NR: not reported, R: right side, SI: slightly improved, SR: spontaneous recovery, SRed: spontaneous reduction, U: unchanged, VF: vocal fold, VHI: voice handicap index.

### 3.4. Quantitative Analysis

The proportion analysis conducted on general anesthesia (GA) groups [14,15,16,17,24,27,28] included a total of seven studies (k = 7), as shown in Figure 2.

The pooled analysis using a random-effects model resulted in an overall effect size of 0.77 (95% CI: 0.62, 0.87; *p* < 0.05). The second proportion analysis (Figure 3) with six studies on local anesthesia (LA) groups [11,12,20,22,25,26] showed an overall effect estimate of 0.89 (95% CI: 0.70, 0.97; *p* < 0.05).

In both proportion analyses, a moderate level of heterogeneity was observed among the included studies (54% and 67%, respectively). The Tau^2^ values of 1.5732 (GA) and 0.4634 (LA) further suggest some variability in effect sizes. The prediction interval, ranging from 0.14 to 1.00 (GA) and from 0.32 to 0.96 (LA), indicates that while the intervention generally appears beneficial, the effect size could vary across different contexts and potentially include no effect. A total of four studies [11,14,20,24] were included in the TT analysis (Figure 4).

The pooled standardized mean difference using a random-effects model was −1.24 (95% CI: −2.20, −0.29), indicating a statistically significant overall effect favouring the experimental group (*p* = 0.03), meaning that a reduced TT positively affected the treatment outcome. Among the individual studies, two showed significant results: Teng 2014 (−2.31, 95% CI: −4.02, −0.59) and Cao 2016 (−1.47, 95% CI: −2.39, −0.56) [11,16]. In contrast, the studies by Rubin 2005 and Hung 2018 did not reach statistical significance, as their confidence intervals were zero [14,24]. Heterogeneity across studies was low, with an I2 value of 11% and a non-significant χ^2^ test for heterogeneity (*p* = 0.34), suggesting that the observed variation in effect sizes was largely due to random errors rather than true differences between studies. The prediction interval, ranging from −2.83 to 0.35, indicates that although the intervention generally favours the experimental group, future studies might observe variability, potentially including null or slightly positive effects.

## 4. Discussion

AS and AD are often underestimated or misdiagnosed conditions which result in the reduced mobility of the vocal fold and incomplete glottic closure [8]. In this updated systematic review and meta-analysis, 13 studies were included. They focused on the diagnosis, treatment, and TT of AS/AD. Nonetheless, the absence of randomized controlled trials (RCTs) limited this investigation. Moreover, a low quality of available evidence of the selected papers was found: no studies scored more than five out of eight for the NOS score.

Revealing a mean patient age of 46.95 years and a predominance of males (53.7%), the demographic data aligned with the knowledge that AS/AD can affect adults across a wide age range, even though pediatric cases have been reported [29]. The prevalence of iatrogenic trauma, specifically due to intubation, as the most common etiology of AS/AD (ranging from 61.76% to 100% of cases) underscores the clinical challenge of managing airways, especially in critical care or surgical settings. External laryngeal blunt trauma, the second cause, pointed to the vulnerability of the laryngeal apparatus due to physical injury and may partially justify the slightly higher prevalence in males [29]. The presence of spontaneous cases, although few, along with idiopathic causes, highlights the complexity of AS/AD’s etiology and the need for thorough diagnostic processes. A notable finding was the higher prevalence of left-sided dislocation (50.1%) over right-sided dislocation (31.6%), with bilateral dislocations being uncommon (2.2%). This asymmetry could suggest an anatomical or biomechanical predisposition for left-sided injuries, possibly related to the direction of endotracheal tube insertion [30]. However, the absence of a standardized classification system complicated the interpretation and comparison of findings across studies, thus limiting our understanding of the condition’s nuances and best treatment approaches. The overlapping symptomatology and endoscopic findings between AS/AD and Recurrent Laryngeal Nerve neuropathies can make differential diagnosis a challenge in the absence of adequate professional expertise and complete diagnostic workouts. The use of fibroscopy and computerized tomography scans for AS/AD identification in the included studies, as well as LEMG in some cases, illustrated the multi-modal approach required, in line with existing diagnostic algorithms [5]. Moreover, the assessment of voice quality through the VHI, GRBAS scale, and acoustic voice analysis provided a comprehensive evaluation framework that underscored the need for specialized care and resources to treat AS/AD patients. Given that ultrasound has recently proven to be a valuable non-invasive imaging technique for detecting the abnormal positioning of the cricoarytenoid joint, there is growing interest in future research on its role in AS/AD, which, to date, has been explored in only a single case report [31].

Concerning treatment, all clinical research groups reported CR as a first-choice treatment, performed with different anesthesiological and surgical techniques. In CR under general anesthesia, according to Rubin et al. (2005), anterior dislocations are reduced using a Hollinger laryngoscope, which facilitates direct visualization and access to anterior laryngeal structures, while posterior dislocations are managed with a Miller-3 laryngoscope positioned in the ipsilateral pyriform sinus, providing appropriate leverage and access for a reduction in posteriorly displaced structures [14]. Subsequently, Leelamanit et al. (2012) introduced a reduction technique using two custom-made stainless-steel rods approximately 30 cm long with a 0.3 cm shaft diameter, tapering to a blunt, hooked tip [16]. According to the authors, this method allows for precise manipulation—with one rod stabilizing the vocal process and the other providing rotational force—which is repeatable and tailored to the dislocation’s subtype that may be preferable to traditional methods, particularly in resistant cases [16]. According to Cao et al. (2016), CR under local anesthesia begins with the application of a 1% tetracaine spray to the oropharyngeal region, followed by further anesthesia directly to the laryngeal surface via an indirect laryngoscope. With the patient phonating the vowel sound “[i:]”, the surgeon uses laryngeal forceps to grasp the posterior–lateral surface of the dislocated arytenoid cartilage and applies gentle rotational force in an antero-upward direction to guide the arytenoid back into position [20]. It was also observed that CR under local anesthesia usually had to be repeated to achieve reduction [20,22,26]. However, there is still controversy regarding the optimal timing for performing closed reduction in order to obtain the best treatment outcome. Previously, relatively stable treatment outcomes and short recovery duration were shown in patients treated between the 13th and 26th day after AD occurrence, with a TT of less than 21 days that seemed to be predictive of the final outcome. According to Lou et al. (2017) [22], after this time, patients may be refractory, and even if a second treatment could be proposed, recovery may only be partial, probably due to the fibrosis caused by mechanical injury that increases the joint’s stiffness, affecting the therapeutic effects of the CR [12]. Sataloff et al. [32] found stable and good treatment outcomes performing CR within 10 weeks after AS or AD, although, in some cases, treatments performed years after the dislocation were also effective [14]. Our investigation revealed that the time from diagnosis to treatment ranged between 1 and 6223 days, with a mean of 80.9 days (SD 122.9). It emerged that a relevant percentage of patients (68.1%) gained full recovery, with their voices returning to their pre-injury quality with affected vocal folds moving normally. Patients who were refractory or felt embarrassed by their voices underwent secondary surgery [14,15,20,27]. In these cases, vocal fold injection was preferred to thyroplasty and voice rehabilitation therapy, as shown in Table 1. Consequently, an early, closed-reduction treatment might improve the success rate and avoid secondary interventions.

The quantitative analysis of the reports supported the qualitative evidence mentioned above. In particular, these findings indicated a high success rate with CR under both general (success rate 77%, CI from 62% to 87%) and local anesthesia (success rate 89%, CI from 70% to 97%). The TT analysis revealed a significant association between reduced time intervals and improved outcomes in AD/AS management. With the majority of standardized mean differences being negative across the four included studies, the estimated average standardized mean difference of −1.24 underscores the clinical significance of timely intervention. These findings emphasize the importance of prioritizing early treatment initiation to optimize patient outcomes and minimize the potential complications associated with delayed intervention. However, further research is warranted to elucidate the optimal timing for treatment initiation and to better understand the underlying mechanisms driving this observed association. None of the analyses showed significant asymmetry in the funnel plot, and the studies included seemed to be reliable as there were no outliers or influential studies.

This systematic review and meta-analysis, while providing valuable insights into the treatment of AS and AD, were characterized by some limitations. The exclusion of non-English language studies may have introduced selection bias, potentially omitting relevant findings published in other languages, while a recent meta-epidemiological study pointed out that the omission of non-English research in the language literature did not significantly change the size or direction of effect estimates or statistical significance in systematic reviews and meta-analysis [33]. Furthermore, the choice of focused search terms, specifically “arytenoid dislocation” and “arytenoid subluxation”, was intended to ensure the relevance and specificity of the included studies. However, this conservative strategy might have excluded studies addressing related conditions (e.g., vocal fold immobility) that could provide indirect insights into AS/AD management. Moreover, the absence of RCTs and the overall moderate-to-low quality of the studies (NOS 5/8) highlighted the importance of the findings’ careful interpretation. Finally, the variability among the studies underscores the crucial need for high-quality, prospective research on this topic. This heterogeneity, while reflecting real-world clinical practice, complicates the process of drawing firm conclusions about the optimal timing and approach for treating AS/AD. Although our analysis supported the hypothesis that a reduced TT interval may be beneficial, the variability in outcomes across the studies suggested that individual patient factors and specific treatment contexts could play crucial roles in determining success. Despite these limitations, this review has several strengths that contribute to its value in the field. It represents one of the most comprehensive efforts to synthesize the available evidence about AS/AD treatment, incorporating a wide range of studies and treatment outcomes. The methodological rigour of our systematic review protocol, adherence to PRISMA guidelines, and the use of the NOS for quality assessment ensure that our analysis is systematic and reproducible. This systematic review and meta-analysis improve our understanding of the optimal timing and treatment modalities for AD/AS. It offers a foundation for future research and clinical practice, emphasizing the importance of early and tailored interventions.

## 5. Conclusions

According to the available evidence, CR appeared to be the preferred treatment for AS/AD, with a high success rate under both general and local anesthesia. A reduced TT seemed to significantly impact recovery from the disease and could be considered a prognostic factor for treatment success. However, given the intrinsic limitations of the available studies on AS/AD, as discussed above, the results of this meta-analysis should not be considered conclusive. Specifically, further data from prospective clinical trials, possibly in a multi-centric setting, are advocated to obtain stronger evidence on the treatment outcomes of AS/AD.

## Figures and Tables

**Figure 1 medicina-61-00092-f001:**
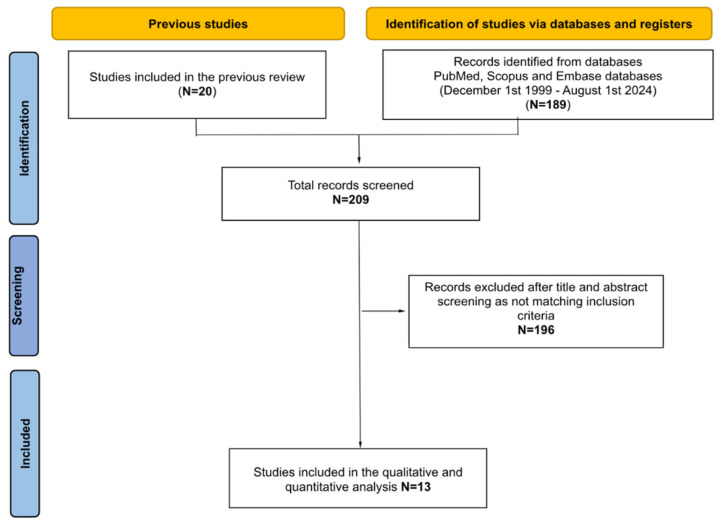
PRISMA diagram from identification to inclusion.

**Figure 2 medicina-61-00092-f002:**
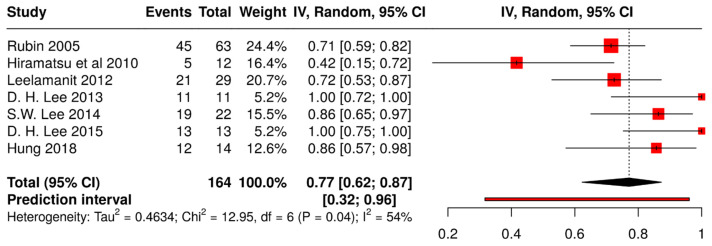
Forest plot showing pooled outcomes of closed reduction under general anesthesia [14,15,16,17,24,27,28].

**Figure 3 medicina-61-00092-f003:**
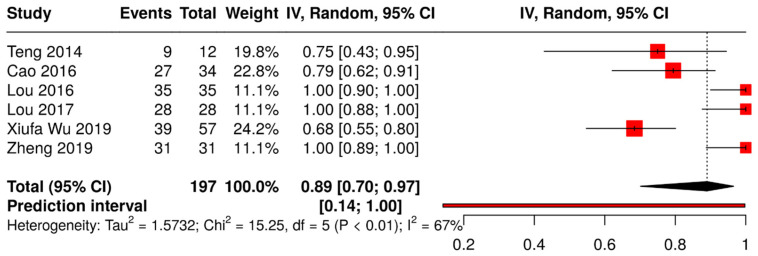
Forest plot showing pooled outcomes of closed reduction under local anesthesia [11,12,20,22,25,26].

**Figure 4 medicina-61-00092-f004:**
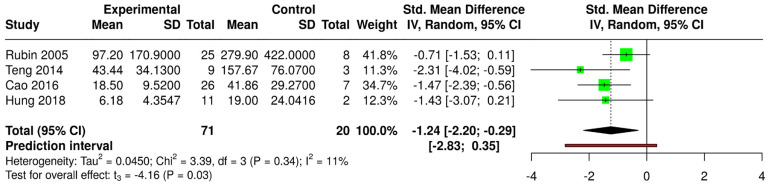
Forest plot showing pooled outcomes of effects of TT on the efficacy of closed reduction [11,14,20,24].

## Data Availability

Data will be made available on request by the corresponding author.

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
