# Peer review of "Optimal Timing and Treatment Modalities of Arytenoid Dislocation and Subluxation: A Meta-Analysis"

_medicina, 2025, doi:10.3390/medicina61010092_

Round 1

Reviewer 1 Report

Comments and Suggestions for Authors

A fascinating subject, a pathology less often encountered in practice and therefore with a less consistent number of studies. For this reason, the bibliography (otherwise suggestive and well chosen) does not contain a very large number of recent articles (from the last 5 years). Correct processing of selected data, correctly and logically expressed. The discussions are broad, pertinent and well oriented on the chosen topic.

Author Response

Reviewer 1

  • A fascinating subject, a pathology less often encountered in practice and therefore with a less consistent number of studies. For this reason, the bibliography (otherwise suggestive and well chosen) does not contain a very large number of recent articles (from the last 5 years). Correct processing of selected data, correctly and logically expressed. The discussions are broad, pertinent and well oriented on the chosen topic.

★Thank you for your comment.

Reviewer 2 Report

Comments and Suggestions for Authors

This paper explores the diagnostic challenges and treatment strategies for arytenoid dislocation and subluxation, focusing on the impact of time to treat on recovery outcomes. The authors aim to synthesize existing data to identify optimal timming and techniques for improving patient outcome.

Abstract. Concise summary of the objectives, methods, and results.

The abstract could benefit from more explicit mention of the limitations of the study. Currently, the lack of randomized controlled trials is mentioned in the conclusion but should also be highlighted here.

Introduction.

The introduction lacks a global epidemiological context. How common is this issue in various populations, and what is the global burden of disease related to AS/AD?

Materials and Methods

Eligibility Criteria: The exclusion of non-English studies might introduce selection bias, as relevant findings in other languages are ignored.

The search terms appear limited to "arytenoid dislocation" and "arytenoid subluxation." Expanding these terms (vocal fold immobility), might have captured additional studies.

Results

No comments

Discussions

No comments

Conclusions

The conclusions overstate the clinical significance of the findings given the limitations of the included studies (small sample sizes, retrospective designs, and lack of RCTs).

References

Some references are outdated (studies from the late 1990s).

Author Response

Reviewer 2

  • This paper explores the diagnostic challenges and treatment strategies for arytenoid dislocation and subluxation, focusing on the impact of time to treat on recovery outcomes. The authors aim to synthesize existing data to identify optimal timing and techniques for improving patient outcome.

Abstract. Concise summary of the objectives, methods, and results. The abstract could benefit from more explicit mention of the limitations of the study. Currently, the lack of randomized controlled trials is mentioned in the conclusion but should also be highlighted here.

★The Abstract has been changed according to Reviewer 2 suggestion.

  • Introduction. The introduction lacks a global epidemiological context. How common is this issue in various populations, and what is the global burden of disease related to AS/AD?

★Thank you for highlighting the need for a more detailed epidemiological context. We have incorporated data from our previous review and additional sources to provide a comprehensive overview of the prevalence and etiology of AS/AD. Specifically, we have discussed the most frequent etiologies and the role of potential risk factors. This additional context aims to enhance the reader's understanding of the global burden of AS/AD.

  • Materials and Methods Eligibility Criteria: The exclusion of non-English studies might introduce selection bias, as relevant findings in other languages are ignored. The search terms appear limited to "arytenoid dislocation" and "arytenoid subluxation." Expanding these terms (vocal fold immobility), might have captured additional studies.

★We appreciate the Reviewer’s concern regarding the exclusion of non-English studies potentially introducing selection bias. This decision was based on practical limitations in translating non-English articles and ensuring the reliability of data extraction. While this may exclude some relevant studies, previous systematic reviews have demonstrated that the majority of high-impact, clinically significant research is published in English [Nussbaumer-Streit, B., Klerings, I., Dobrescu, A. I., Persad, E., Stevens, A., Garritty, C., Kamel, C., Affengruber, L., King, V. J., & Gartlehner, G. (2020). Excluding non-English publications from evidence-syntheses did not change conclusions: a meta-epidemiological study. Journal of clinical epidemiology, 118, 42–54. https://doi.org/10.1016/j.jclinepi.2019.10.011]. Regarding the specificity of the search terms, we agree that expanding search terms to include broader terms such as "vocal fold immobility" might have captured additional studies. However, our focus was specifically on arytenoid dislocation and subluxation, which have distinct diagnostic and management pathways compared to other causes of vocal fold immobility, and the search terms used adhere to precedent review on the theme (Frosolini, 2020; Norris, 2011). Using broader terms could have diluted the search results with studies unrelated to AS/AD. We chose to prioritize specificity to ensure that the included studies directly addressed the objectives of this systematic review and meta-analysis. Nevertheless, we recognize the value of broader search strategies and have acknowledged this as a limitation in the discussion.

  • Results: No comments. Discussions: No comments. Conclusions The conclusions overstate the clinical significance of the findings given the limitations of the included studies (small sample sizes, retrospective designs, and lack of RCTs).

★According to Reviewer 2 suggestions, the conclusions were changed as follows: “According to the available evidence, the diagnostic workup for AS/AD should include endoscopy, radiologic imaging and LEMG. CR appears to be the preferred treatment, with a high success rate under both general and local anesthesia. A reduced TT seems to significantly impacts recovery from the disease and can be considered a prognostic factor for treatment success. However, given the intrinsic limitations of the available studies on AS/AD, as discussed above, the results of this metanalysis should not be considered as conclusive. Specifically, further data from prospective clinical trials, possibly in a multi-centric setting, are advocated to obtain stronger evidence on treatment outcomes of AS/AD.”

  • References. Some references are outdated (studies from the late 1990s).

★We appreciate the reviewer’s observation regarding the inclusion of older references in our manuscript. While some studies from the late 1990s were cited, these references were carefully selected due to their foundational contributions to the understanding of arytenoid dislocation and subluxation, particularly in terms of patho-physiology and early treatment techniques. These earlier works provide essential context and historical perspective that underpin subsequent advancements in the field. Moreover, while we have included recent publications where available, the limited volume of new research on this topic necessitates the use of older references to ensure a comprehensive and accurate representation of the existing evidence. 

Reviewer 3 Report

Comments and Suggestions for Authors

Journal: Medicina (ISSN 1648-9144)

Manuscript ID: medicina-3396532

Type: Systematic Review

Title: Optimal timing and treatment modalities of arytenoid dislocation and subluxation: a meta-analysis.

Authors

Andrea Frosolini , Valeria Caragli , Giulio Badin , Leonardo Franz , Patrizia Bartolotta , Andrea Lovato , Luca Vedovelli , Elisabetta Genovese , Cosimo de Filippis , Gino Marioni

 I appreciate the opportunity to review this interesting paper.

This document provides a context for a systematic review and meta-analysis on managing surgical techniques for arytenoid dislocation (AD) and subluxation (AS). The prevalence of arytenoid dislocation in tracheal intubation during surgery or after external trauma is notably high; however, it has not garnered adequate attention.

The primary objective of this systematic review and meta-analysis was to present treatment options for closed reduction (CR) under local and general anesthesia. The secondary objective was to investigate the importance of the time between diagnosis and surgical treatment and its impact on the patient's voice.

The authors have put great effort into analyzing the available literature, resulting in a comprehensive review of the articles.

The authors identified the research gap as a lack of high-quality evidence, particularly from randomized controlled trials, in the existing literature on AD/AS treatment methods and the best time for providing surgery. A similar review was published in 2020 partly by the same authors, which is one of the few caveats to this article, but I found the topic still interesting (Frosolini, A.; Marioni, G.; Maiolino, L.; de Filippis, C.Lovato A. Current management of arytenoid subluxation and dislocation). Eur. Arch. Otorhinolaryngol. 2020, 277, 2977–2986. doi:10.1007/s00405-020-06042-3).

 The work uses citations from 1998 to 2024, but there are few new citations after 2020, which is a weakness (seven citations from 2020 to 2024). This study's advantage is that it possesses only one self-citation.

 The Introduction section provides general information on the anatomy of the cricoarytenoid joints and treatment methods.

The Materials and Methods section describes the research question, the inclusion and exclusion criteria, and the data analysis methods. The study protocol was registered in PROSPERO (study ID: CRD42023407521).

The final part of the PRISMA diagram in the results section of Figure 1 is unclear.

Why did the authors include 13 and subsequently 12 studies, respectively?

Are some of the studies in Table 1 that used other treatment methods, such as laryngoplasty, thyroplasty, or botulinum toxin injections, excluded from the analysis? If not, what impact did the additional methods have on the treatment results?

This discussion is both extensive and interesting. The conclusions should be corrected because the work did not analyze diagnostic methods ("The diagnostic workup for AS/AD should include endoscopy, radiologic imaging, and LEMG").

The authors' mention of the study's limitations, the lack of randomized studies, and the coexistence of other treatment methods does not allow for an unequivocal conclusion about the equivalence of local and general anesthesia in treating AS/AD.

Nevertheless, choosing the anesthesia method and deciding the time of the procedure are essential for a practicing physician.

Author Response

Reviewer 3

  • Thisdocument provides a context for a systematic review and meta-analysis on managing surgical techniques for arytenoid dislocation (AD) and subluxation (AS). The prevalence of arytenoid dislocation in tracheal intubation during surgery or after external trauma is notably high; however, it has not garnered adequate attention. The primary objective of this systematic review and meta-analysis was to present treatment options for closed reduction (CR) under local and general anesthesia. The secondary objective was to investigate the importance of the time between diagnosis and surgical treatment and its impact on the patient's voice. The authors have put great effort into analyzing the available literature, resulting in a comprehensive review of the articles.

The authors identified the research gap as a lack of high-quality evidence, particularly from randomized controlled trials, in the existing literature on AD/AS treatment methods and the best time for providing surgery. A similar review was published in 2020 partly by the same authors, which is one of the few caveats to this article, but I found the topic still interesting (Frosolini, A.; Marioni, G.; Maiolino, L.; de Filippis, C.Lovato A. Current management of arytenoid subluxation and dislocation). Eur. Arch. Otorhinolaryngol. 2020, 277, 2977–2986. doi:10.1007/s00405-020-06042-3). 

★Thanks for your comment.

  • The work uses citations from 1998 to 2024, but there are few new citations after 2020, which is a weakness (seven citations from 2020 to 2024). This study's advantage is that it possesses only one self-citation.

★Thanks for your observation regarding the range of citations used in this study, as highlighted by Reviewer 2. While it is true that there are relatively few new citations after 2020, this reflects the limited volume of recent literature specifically addressing arytenoid dislocation and subluxation. The field remains underexplored, with foundational and seminal studies from earlier years continuing to provide critical insights into the patho-physiology, diagnosis, and treatment of these conditions. Where recent literature is available, we have incorporated it into the manuscript: these newer references highlight advancements and emerging hypotheses, ensuring our investigation reflects the current state of knowledge. 

  • The Introduction section provides general information on the anatomy of the cricoarytenoid joints and treatment methods. The Materials and Methods section describes the research question, the inclusion and exclusion criteria, and the data analysis methods. The study protocol was registered in PROSPERO (study ID: CRD42023407521). The final part of the PRISMA diagram in the results section of Figure 1 is unclear. Why did the authors include 13 and subsequently 12 studies, respectively?

★Thank you for your comment. It was a typo: 13 articles have been included both in our qualitative and quantitative analysis. We have now uploaded the correct version of the PRISMA table. 

  • Are some of the studies in Table 1 that used other treatment methods, such as laryngoplasty, thyroplasty, or botulinum toxin injections, excluded from the analysis? If not, what impact did the additional methods have on the treatment results?

★Thanks for your comment. Table 1 reports both primary treatment (closed reduction) and, in some cases, adjunctive modalities such as laryngoplasty, thyroplasty, and botulinum toxin injections that were performed in a limited number of cases: 33 patients out of 361 patients [14,15,20,27]. These additional interventions were not excluded from our analysis but were integrated to provide an understanding of their role in managing arytenoid dislocation and subluxation. The inclusion of adjunctive treatments in the studies analyzed does not undermine the efficacy of CR as the primary intervention but rather demonstrates the need for tailored approaches in select cases: procedures as laryngoplasty, thyroplasty, and botulinum toxin injections are valuable in addressing specific challenges, such as incomplete arytenoid repositioning, persistent dysphonia, or muscle imbalance. These treatments enhance patient outcomes, particularly in complex or delayed cases, as we already discussed in the manuscript (see lines 325-328). Thanks for pointing out this aspect, we incorporated aggregated data in the results of this revised version of the manuscript (lines 206-208).

  • This discussion is both extensive and interesting. The conclusions should be corrected because the work did not analyze diagnostic methods ("The diagnostic workup for AS/AD should include endoscopy, radiologic imaging, and LEMG"). The authors' mention of the study's limitations, the lack of randomized studies, and the coexistence of other treatment methods does not allow for an unequivocal conclusion about the equivalence of local and general anesthesia in treating AS/AD. Nevertheless, choosing the anesthesia method and deciding the time of the procedure are essential for a practicing physician.

★Thank you for your thoughtful comment. We recognize that our Conclusions section included a discussion of diagnostic methods that were not specifically analyzed in this study. This oversight has been addressed by refining the Conclusions section to focus exclusively on the study's findings related to treatment modalities and timing.